



1  **A European map of groundwater pH and calcium**

3  Michal Hájek[1*], Borja Jiménez-Alfaro[1,2*], Ondřej Hájek[1], Lisa Brancaleoni[3], Marco Cantonati[4],

4  Michele Carbognani[5], Anita Dedić[6],  Daniel Dítě[1,7], Renato Gerdol[3], Petra Hájková[1], Veronika

5  Horsáková[1], Florian Jansen[8], Jasmina Kamberović[9], Jutta Kapfer[10], Tiina Kolari[11], Mariusz

6  Lamentowicz[12], Predrag Lazarević[13], Ermin Mašić[14], Jesper Erenskjold Moeslund[15], Aaron Pérez-

7  Haase[16,17], Tomáš Peterka[1],  Alessandro Petraglia[5], Eulàlia Pladevall-Izard[17], Zuzana Plesková[1],

8  Stefano Segadelli[18], Yuliya Semeniuk[19], Patrícia Singh[1], Anna Šímová[1], Eva Šmerdová[1], Teemu

9  Tahvanainen[11], Marcello Tomaselli[5], Yuliya Vystavna[20,21], Claudia Biţă-Nicolae[22], Michal Horsák[1]

11  *the authors contributed equally to the study

13  [1] Department of Botany and Zoology, Faculty of Science, Masaryk University, Kotlářská 2, 61137

14  Brno, Czech Republic

15  [2] Research Unit of Biodiversity (CSIC/UO/PA), University of Oviedo, Spain

16  [3] Department of Life Sciences and Biotechnology, University of Ferrara, Corso Ercole I d'Este 32,

17  44121 Ferrara, Italy

18  [4] MUSE – Museo delle Scienze, Limnology & Phycology Section, Corso del Lavoro e della Scienza

19  3, Trento, 38123, Italy



[5] Department of Chemistry, Life Sciences and Environmental Sustainability, University of Parma,
Parco Area delle Scienze 11/A, 43124 Parma, Italy
[6] Faculty of Science and Education, University of Mostar, Rodoč bb, 88 000 Mostar, Bosnia and
Herzegovina
[7] Plant Science and Biodiversity Center, Slovak Academy of Sciences, Dúbravská cesta 9, 84523
Bratislava, Slovak Republic
[8] Faculty of Agricultural and Environmental Sciences, University of Rostock, Justus-von-Liebig-
Weg 6, 18059 Rostock, Germany
[9] Faculty of Natural Sciences and Mathematics, University of Tuzla, 75 000, Tuzla, Bosnia and
Herzegovina
[10] Department of Landscape Monitoring, Norwegian Institute of Bioeconomy Research,
Holtvegen 66, 9016 Tromsø, Norway
[11] Department of Environmental and Biological Sciences, University of Eastern Finland, Joensuu
campus, Yliopistokatu 7, 80101 Joensuu, Finland
[12] Climate Change Ecology Research Unit, Faculty of Geographical and Geological Sciences,
Adam Mickiewicz University in Poznan, Bogumiła Krygowskiego 10, 61-680 Poznaň, Poland
[13] Institute of Botany and Botanical Garden "Jevremovac", Faculty of Biology, University of
Belgrade, Takovska 43, 11000 Belgrade, Serbia
[14] Department of Biology, Faculty of Science, University of Sarajevo, Zmaja od Bosne 33-35,
71000 Sarajevo, Bosnia and Herzegovina



[15] Section of Biodiversity, Department of Bioscience, Aarhus University, Grenaavej 14, 8410
Roende, Denmark
[16] Department of Biosciences, Faculty of Sciences and Technology, University of Vic - Central
University of Catalonia, Vic, Spain, EU
[17] Department of Evolutionary Biology, Ecology and Environmental Sciences, Faculty of Biology,
University of Barcelona, Barcelona, Spain
[18] Geological, Seismic and Soil Service, Emilia-Romagna Region, Bologna, Italy
[19] Stanislaw Leszczycki Institute of Geography and Spatial Organization, Polish Academy of
Science, Warsaw, Poland
[20] Institute of Hydrobiology, Biology Centre CAS, Na Sadkach 7, 37005, Ceske Budejovice, Czech
Republic
[21] Isotope Hydrology Section, International Atomic Energy Agency, Vienna, Austria
[22] Institute of Biology Bucharest, Romanian Academy, Bucharest, Romania

**Abstract.** Water resources and associated ecosystems are becoming highly endangered due to
ongoing global environmental changes. Spatial ecological modelling is a widely used tool for
understanding the past, present and future distribution and diversity patterns in groundwater-
dependent ecosystems, such as fens, springs, streams, reed beds or wet grasslands. Still, the
lack of detailed water chemistry maps prevents their reasonable use on continental and global
scales. Being major determinants of biological composition and diversity of groundwater-
dependent ecosystems, groundwater pH and calcium are of utmost importance. Here we
developed the up-to-date European map of groundwater pH and Ca, based on 7,577
measurements of near-surface groundwater pH and calcium distributed across Europe.  In
comparison to the existing European groundwater maps, we included a several times larger
number of sites, especially in the regions rich in spring and fen habitats, and filled the apparent
gaps in Eastern and Southeastern Europe. We used Random Forest models and regression
kriging to create continuous maps of water pH and calcium at the continental scale, which is
freely available also as a raster map (Hájek et al. 2020; 10.5281/zenodo.4139912). Lithology
had higher importance than climate for both pH and calcium. The previously recognised
latitudinal and altitudinal gradients were rediscovered with much refined regional patterns, as
associated with bedrock variation. For ecological models of distribution and diversity of
groundwater-dependent, but also other terrestrial, ecosystems, the new map is more suitable
than previously used maps of soil pH, unlike which it mirrors bedrock chemistry more than
vegetation-dependent soil processes.




**1.  Introduction**
The Earth system is currently undergoing unprecedented changes in climate, global
biogeochemical cycles, and land use, resulting in biodiversity loss (Ceballos et al. 2017, Song et



al. 2018, Blowes et al. 2019, Brondizio et al. 2019). Freshwater systems belong to the most
endangered habitats (Cantonati et al. 2020a, Tickner et al. 2020) and, among them,
groundwater-dependent ecosystems, such as fens and springs, hold primacy (Janssen et al.
2016, Chytrý et al. 2019, Hájek et al. 2020). Species composition and richness of spring systems
are generally governed by water pH and calcium concentration ($Ca^{2+}$), which are highly variable
at different spatial scales (Malmer 1986; Rydin and Jeglum 2013; Peterka et al. 2017; Horsáková
et al. 2018; Cantonati et al. 2020a,b). Therefore, understanding the spatial patterns in
groundwater pH and $Ca^{2+}$ is important not only for general geochemical knowledge and for
water resource management, but to the same extent for the conservation of freshwater
systems and associated biodiversity.

In Earth and biodiversity sciences, ecological modelling is a widely used tool for

understanding the distribution and diversity patterns of ecosystems and habitats, and for
predicting their future development under global change. Ecological models usually incorporate
environmental or historical predictors extracted from thematic maps (Jiménez-Alfaro et al.
2018a, Večeřa et al. 2019, Divíšek et al. 2020), including soil properties for terrestrial
ecosystems (Hengl et al. 2017). However, soil parameters as soil pH contribute negligibly to the
models for groundwater-dependent habitats, even for those strongly controlled by pH and $Ca^{2+}$,
such as base-rich fens (Jiménez-Alfaro et al. 2018b). This is due to a poor correlation between
groundwater chemistry and pH or $Ca^{2+}$ in soil, disrupted mainly by mineral leaching or
accumulation of organic matter in soil. For this reason, there is a strong need to produce maps
for groundwater pH and $Ca^{2+}$ concentration at the European scale that would allow producing



the continental-scale ecological models useful for enforcing conservation strategies in
groundwater-dependent habitats.

In spite of important mapping efforts of groundwater (Duscher et al. 2015) and karst

aquifers (Chen et al. 2017) at the European and global level, the only available European-scale
maps of groundwater pH and $Ca^{2+}$ concentration are those included in the FOREGS Geochemical
Atlas of Europe (Salminen et al. 2006). These maps are based on 808 stream-water
measurements distributed relatively equally across Europe. However, they show a large gap in
Eastern and Southeastern Europe (Romania, Bulgaria, Belarus, Russian Federation, Ukraine,
Moldova, Serbia, Kosovo, Montenegro, Bosnia and Herzegovina, Northern Macedonia). In
addition, the maps are constructed based on insufficient data density in some areas rich in)
different groundwater-dependent ecosystems, but heterogeneous in terms of lithology (the
Alps, the Carpathians, Bohemian Massif, the Cantabrian Mountains and the Pyrenees, and some
regions of Fennoscandia). We therefore aimed at substantial improvement of the existing data
by creating a database with field data measurements across the entire European continent, and
at creating a model-based map representing major patterns of groundwater pH and $Ca^{2+}$
concentration at local and continental scales. Our data will allow better understanding of the
patterns and causes of groundwater conditions in freshwater systems, and strongly improve
the databases for European-scale modelling of the biodiversity in groundwater-dependent and
related ecosystems.

**2.  Methods**





*Data collection*
We assembled the data set of pH and $Ca^{2+}$ (or electrical conductivity in $\mu S.cm^{-1}$ at 20 °C;
hereinafter abbreviated as EC) measurements in groundwater, covering the whole of Europe,
with a greater density in the regions rich in endangered groundwater-dependent ecosystems
such as springs and fens. We excluded most of Ukraine and European part of Russian
Federation, because of large data gaps in these areas. We considered all types of shallow
groundwater systems, especially spring, spring-fen, and stream water. The core of our data set
is formed by unpublished pH and $Ca^{2+}$ or EC data sets of co-authors (3,618 sites); some of them
processed in ecological papers without presenting original pH and $Ca^{2+}$ data (Hájková et al. 2006,
2008; Hájek et al. 2008; Sekulová et al. 2013; Plesková et al. 2016; Horsáková et al. 2018;
Šímová et al. 2019). The second most important source were vegetation databases registered in
GIVD (Dengler et al. 2011; Table 1) and EVA (Chytrý et al. 2016), from where 1,160
measurements from freshwater habitats were obtained. Both unpublished data and data from
vegetation databases were filtered using original information or metadata of the sources in a
way that only data from spring-fed fens and springs were considered. The data from
ombrotrophic bogs and clearly topogenic fens (mainly terrestrialised lakes) were omitted
because their water chemistry is governed by the decomposition of organic matter,
atmospheric humidity and deposition, algal photosynthesis (Kann and Smith 1999), and biotic
processes such as cation exchange capacity of mosses (Clymo 1963, Soudzilovskaia et al. 2010,
Vicherová et al. 2015), rather than by bedrock chemistry. We also obtained data from public
data sets from national environmental and nature conservation agencies of Germany, Slovenia
and Bulgaria (1081 sites; see Table 1), data from FOREGS Geochemical Atlas of Europe

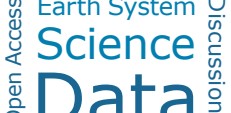

(Salminen et al. 2006; 808 sites) and literature data based on our gap-oriented excerption (883
sites; Table 1); most data came from Hinterlang (1992), Tanneberger et al. (2011), Eades et al.
(2018), Kadūnas et al. (2017) and Savić et al. (2017).

**Table 1.** Data sources. The name abbreviations are explained in the team list or
acknowledgements.





## I.  Vegetation databases.

| name | n | GIVD code | custodian |
|---|---|---|---|
| European Mire Vegetation Database (EMVD) | 510 | GIVD EU-00-022 | T.P. |
| National Vegetation Database of Denmark | 373 | EU-DK-002 | J.E.M. |
| Britain_nvcd | 224 | GIVD EU-GB-001 | J.R. |
| Balkan Vegetation Database | 27 | GIVD EU-00-019 | K.V. |
| Germany_vegmv | 19 | GIVD EU-DE-001 | F.J. |
| Basque country | 7 | EU-00-011 | I.B. |

## II.  Unpublished data sets

| Regions | n | co-authors of the data set |
|---|---|---|
| Central and Eastern Europe | 1405 | Z.P., M.Há., P.H., T.P., D.D., L.S., M.L.,  J.N., P.P., P.S., A.Š., Y.S., C.B.-N. |
| Spain | 645 | A.P-H., E.P-I., B.J-A. |
| Bulgaria | 428 | P.H., M.Há., M.Hor. |
| Fennoscandia | 392 | M.Há., T.P., D.D., M.Hor., V.H., P.H., T.K., T.T., J.Kap., D.-I.Ø. |
| Apennines | 285 | M.T., M.Can., M.Car., S.S., A.P., L.B., R.G. |
| Europe (cross-taxon research) | 281 | M.Há., P.H., D.D., M.Hor., V.H. |
| Balkans except Bulgaria | 134 | A.D., E.M., J.Kam., P.L., T.P., M.Há., P.H. |

## III.  Public data sets

| area and agency | n | provided via |
|---|---|---|
| North Rhine-Westphalia (LANUV agency; D) | 463 | Dr. Dirk Hinterlang, Dr. Sabine Bergmann |
| Ministry of Environment and Water (BG) | 442 | Mrs. Rossitza Gorova |
| Ministry of the Environment (SI) | 176 | http://www.arso.gov.si/, assessed 26 February 2019 |

## IV.  Geochemical Atlas of Europe

| area | n | reference |
|---|---|---|
| Europe | 808 | Salminen et al. 2006 |

## V.  Other literature data (gap-oriented excerption)

| region and context | n | reference |
|---|---|---|
| West-Central European springs | 340 | Hinterlang 1992 |
| Lithuanian springs | 194 | Kadūnas et al. 2017 |
| NE England | 111 | Eades et al. 2018 |
| N Germany | 58 | Tanneberger et al. 2011 |
| Central Bosnia | 50 | Savić et al. 2017 |
| Western Bohemian mineral springs (CZ) | 28 | Laburdová and Hájek 2014 |
| Scotland | 24 | Gorham 1957 |
| Eastern Bosnia | 20 | Kamberović et al. 2019 |



| Switzerland (mires) | 14 | Lamentowicz et al. 2010 |
| British and French travertines | 13 | Pentecost and Zhaohui 2002 |
| NW Poland (mires) | 8 | Lamentowicz and Mitchell 2005 |
| Western Balkans | 6 | Ridl et al. 2018 |
| Kosovo | 5 | Kelmendi et al. 2018 |
| SE Croatia | 4 | Terzić et al. 2014 |
| Kosovo (Rugova) | 3 | Lajçi et al. 2017 |
| Serbia | 3 | Ćirić et al. 2018 |
| NE Croatia | 1 | Špoljar et al. 2011 |
| SE Croatia (Krčić) | 1 | Kolda et al. 2019 |


In total, we collected 7,577 samples (Table 1). A part of the samples, however,
represented repeated measurements conducted in the same site, especially in public data sets.
Some samples from other data sets (vegetation databases, literature data) shared the same
coordinates and site name or code, suggesting repeated measurements as well. We therefore
averaged repeated measurements from the same sampling spots. We further deleted samples
whose coordinates were obviously erroneous, such as those in oceans. These steps reduced the
number of samples to 6,561, out of which 6,459 samples contained information on water pH
value and 5,927 samples contained information about EC of water or $Ca^{2+}$ concentration. Out of
these 5,927 samples, 2,988 samples had directly measured both $Ca^{2+}$ and EC ($\mu S.cm^{-1}$ at 20 °C),
and for the remaining 2,939 samples we estimated $Ca^{2+}$ concentration by EC of water.

*Imputation of missing $Ca^{2+}$ values by EC of water*
For imputation of $Ca^{2+}$ values based on EC, we first aimed at constructing a simple imputation
equation based on the well-known correlation between EC and $Ca^{2+}$ concentration in springs
and fens (Sjörs & Gunnarsson 2002, Plesková et al. 2016). In our data set of 2,988 samples, as
well as in its regional subsets, this relationship was strongly governed by EC values above ca
1,000 µS.cm$^{-1}$, although they formed only a small part of the data set (4.7% of the data set; 139
samples). In the EC range 1,000-10,000 µS.cm$^{-1}$ (an outlier of 17,000 µS.cm$^{-1}$ was omitted), the
correlation between water EC and Ca$^{2+}$ concentration was not statistically significant (r = 0.15, P
= 0.07). The problem of high EC values governing the regression model was the most apparent
in the public data sets. In the data set of Bulgarian Ministry of Environment, weak correlation
between EC and Ca$^{2+}$ persisted even when EC values above 1000 were omitted (Supplementary
Figure 1). This database further contains many samples which are not near-surface samples
that were measured in other datasets. We therefore finally decided (1) not to include the
database of Bulgarian Ministry of Environment into the imputation model, and (2) limit the
gradient of EC to 1,000 µS.cm$^{-1}$. We further omitted a few samples from ophiolite (Kamberović
et al. 2019) where high EC occurred despite low Ca$^{2+}$. The resulting data set of 2,319 samples
nevertheless still showed some samples with suspiciously high or low Ca$^{2+}$ concentration relative
to EC (Supplementary Figure 2),  suggesting either the effect of other ions than Ca$^{2+}$ or
inconsistent analytical methodology. Because our aim was to create the most accurate
imputation model rather than testing the relationship, we removed these outliers. Therefore,
we calculated the EC:Ca and Ca:EC ratios and removed outliers, i.e., all points outside the 1.5 x
interquartile range. The final imputation model was hence based on 2,062 sites.  We performed
a null-intercept linear regression (Figure 1) with Ca$^{2+}$ as dependent variable (y) and EC as
predictor (x); the resulting equation y = 0.153x was obtained (R$^2$ = 0.84). Based on this
equation, we imputed Ca$^{2+}$ concentrations to all samples where only EC was measured. The
imputed Ca$^{2+}$ values show a somewhat narrower range (Supplementary Figure 3) than originally



measured values. Both subsets show minimum $Ca^{2+}$ value below 1 mg.$l^{-1}$, but imputed data
show lower non-outlier maximum (125.5 mg.$l^{-1}$) than measured data (197.2 mg.$l^{-1}$). Absolute
maximum value was also lower for the subset with imputed values. Imputation of $Ca^{2+}$ values to
all samples, including high-EC ones (> 1,000 $\mu S.cm^{-1}$), hence did not skew the imputed data to
higher values.

**Figure 1:** The final regression model to impute $Ca^{2+}$ values based on electrical conductivity (EC;
$\mu S.cm^{-1}$; n = 2,062

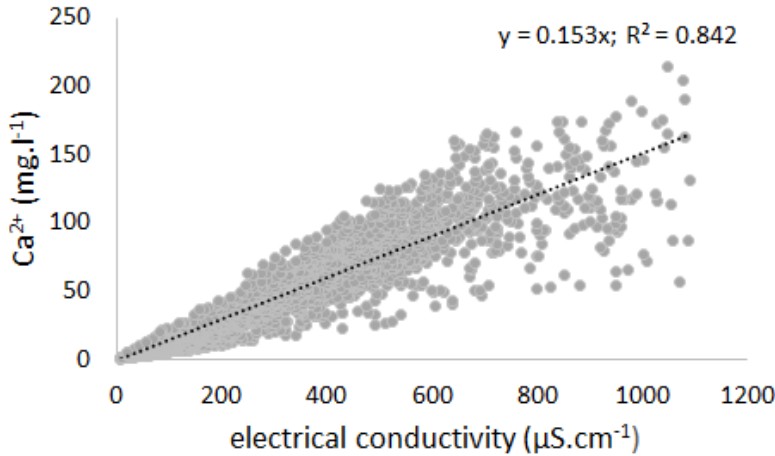


*Geographical modelling*
We used our dataset with 6,561 sites with measured groundwater pH and either measured or
imputed $Ca^{2+}$ concentrations to model expected values across non-sampled areas.
Groundwater-dependent pH (GW-pH) was modeled with 6,459 samples (pH min = 2.20; max =



11.32; mean = 6.69); and groundwater-dependent $Ca^{2+}$ (GW-Ca) with 5,927 samples (min =
0.15; max = 3567.41; mean = 48.73 mg.l$^{-1}$). $Ca^{2+}$ values were ln-transformed. All field samples
had geographic coordinates assigned from GPS or georeferenced with an accuracy between ca
10 (precise field measurements) and 500 m (some database data). We kept the pH outliers; ten
values below 3.5 and nine values above 8.8. Even if these values may be suspicious, they largely
come from published sources (FOREGS Geochemical Atlas of Europe, British vegetation
database). Apart from measurement error, they may be explained by the influence of mineral
waters from deep hydrological circulations (e.g., in a spring in the Apennines, very high pH
value 11.2 was due to enrichment with sodium and chloride associated with low temperature
reaction between meteoric water and ultramafic rocks; Boschetti and Toscani 2008, Boschetti
et al. 2013, Segadelli et al. 2017, Cantonati et al. 2020c). These values form only a minor part  of
the data set and have negligible effect on the results.

For each site, we obtained environmental predictors from the thematic GIS maps (see

below). For some sites, important predictors were missing in the maps (e.g., sites at far north in
the arctic zone, or close to sea or water bodies) and these sites were therefore not included in
the final models.

*Numerical analyses*
Numerical analyses were done in R version 3.6.3 (R Core Team, 2020), with the support of
ArcGIS 10.2 (ESRI, Redlands, CA) for geoprocessing and map production. We first conducted
exploratory analyses to test the Ca and pH prediction ability of different GIS layers related to



soil bedrock, climate, and topography. We focused on layers with a complete coverage of
Europe with an eastern border from the Black Sea in Turkey to the White Sea in Russian
Federation, thus including the regions with a relatively good cover of field measurements
(Figure 2). We performed Linear Models for individual variables to select those providing
significant relationships and > 1% of explained variance. A variable for soil pH (measured in
water solution) at 15 cm depth for a 250 m grid resolution provided by the soilgrids project
(www.soilgrids.org) had the highest explanatory power for GW-pH ($R^2$ = 0.22) and GW-Ca ($R^2$ =
0.16). The same results were obtained when using the same variable for 5 or 10 cm depth. We
also tested soil estimates from Ballabio et al. (2019), but they provided weaker relationships for
both GW-pH ($R^2$ = 0.14 using soil pH as a predictor) and GW-Ca ($R^2$ = 0.01 using soil pH, $R^2$ = 0.01
using soil $CaCO_3$). To account for lithology, we used the lithological groups (litho3 level)
included in the polygon layer of the Hydrogeological map of Europe (Duscher et al. 2015) as a
categorical variable. We also selected annual precipitation (Bio12) as provided in CHELSA
(Karger et al. 2017) to account for precipitation gradients which are expected to influence
groundwater regimes. Other variables related to precipitation were highly correlated with
annual precipitation (Pearson r > 0.75) and omitted. The variables of lithology and soil pH were
aggregated to the same grid extent of CHELSA at 1 km resolution, using the dominant unit and a
bilinear interpolation, respectively.

**Figure 2.** Spatial distribution of the three groups of calibration data collected for modelling
groundwater pH and $Ca^{2+}$ in European fens (original and literature data from springs and fens;
data from streams from FOREGS Geochemical Atlas of Europe; other data). Other data include



public data from national groundwater monitoring of Bulgaria and Slovenia. For separate maps
of pH and $Ca^{2+}$ see Supplementary Figure 5.

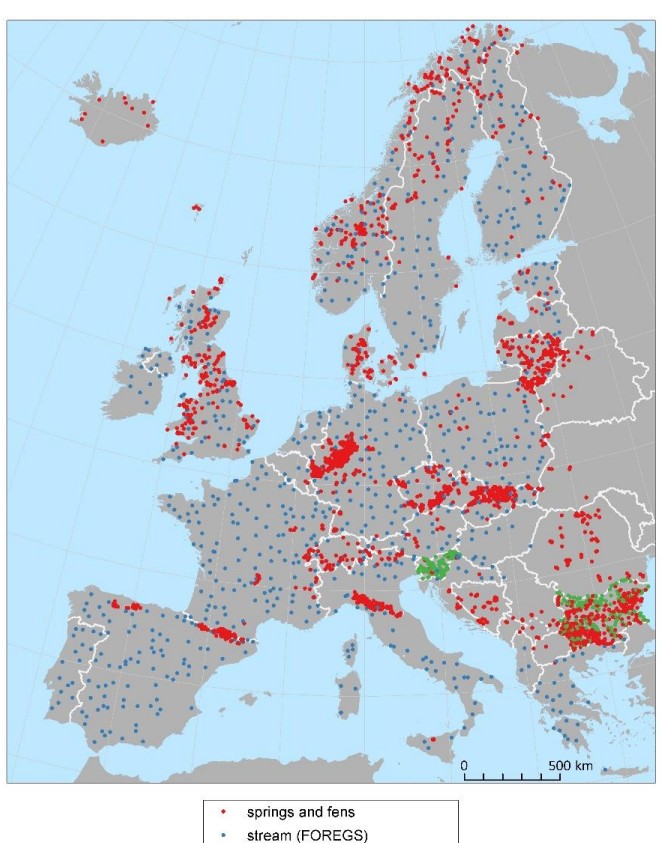



Spatial predictions were based on regression kriging (RK), a technique that combines a
regression model based on explanatory variables with the interpolation of model residuals with
ordinary kriging (Hengl et al. 2007; Meng et al. 2013). RK is especially appropriate for modeling
soil attributes at medium and large scales, combining the spatial autocorrelation of soil



variables with the explanatory power of auxiliary variables (Keskin and Grunwald 2018). We
implemented RK with the GSIF R package (Hengl 2020). As the regression component, we
computed Random Forests since a preliminary analysis with our data showed better
performance than linear models, generalised linear models, or generalised additive models.
Random Forests are ensemble learning methods based on decision trees and an internal
correction of overfitting, which provide high interpretability and good performance when
compared with other algorithms used in soil spatial modeling (Wiesmeier et al. 2011). Another
advantage of Random Forests is that they have no requirements for considering the probability
distribution of soil variables, fitting complex non-linear relationships for spatial extrapolation
(Hengl et al. 2015). We fitted the Random Forests model and the residual variogram for
groundwater pH and Ca$^{2+}$ separately using the function fit.gstatModel() in GSIF package. Spatial
predictions were then computed with the predict() function using the model object generated
previously and a 5-fold cross-validation. Model evaluation was based on the calculation of the
Mean Error (ME) and the Root Mean Squared Error (RMSE) as the differences between
predicted and observed values (Keskin and Grunwald 2018; Pham et al. 2019). We compared
the relationships between the models for groundwater pH and Ca$^{2+}$ by using a random sampling
of 5,000 points to extract cell values and computing a Pearson correlation. To assess regional
differences, we correlated values grouped in 25 neighboring cells of each single cell using the
rasterCorrelation() function in the package spatialEco.

**3.  Results**



In measured data, ranges and medians of pH and $Ca^{2+}$ concentration was similar across
Europe (Supplementary Figure 4), with lowest pH values found in the Atlantic and Iberian
regions and highest pH values found in southern Europe except Iberian Peninsula. Lowest $Ca^{2+}$
values were found in boreal Europe, while the highest in Central and Southern Europe. The
Random Forest models computed with the lithology, soil pH, and precipitation explained 40%
and 55% of the variance for GW-pH and GW-Ca, respectively. Lithology was the variable with
the highest importance in both models (Figure 3), although its effect was higher in the model
computed for $Ca^{2+}$ than for pH. Conversely, soil pH had higher relative importance in GW-pH
than GW-Ca, while precipitation had the lowest contributions in both models.

**Figure 3**. Variable importance of Random Forest models computed for groundwater pH and
$Ca^{2+}$. MSE = Mean Standard Error.

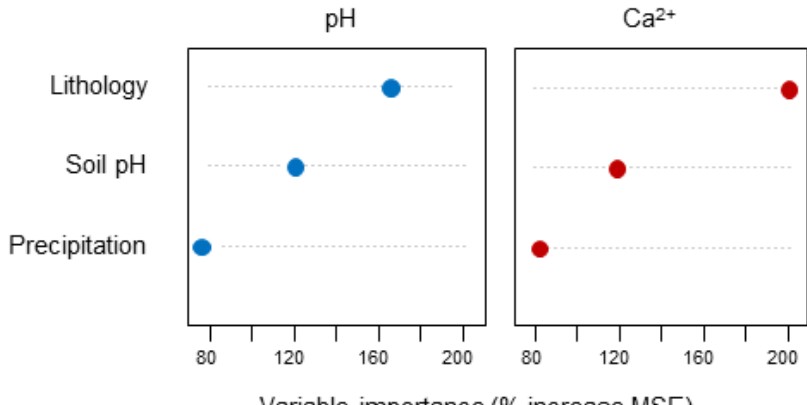








When adding the kriging component, model predictions reached 65% and 74% of explained
variance for GW-pH and GW-Ca, respectively. The mean values of standard errors (SE; 0.00058
for pH; -0.0009 for Ca) and Root Mean Squared Errors (RMSE; 0.588 for pH; 0.690 for Ca) were
higher in the models for pH, but in both cases showed low values and accurate predictions, in
agreement with their total explained variance.
Model predictions for groundwater pH reflected the lowest values in Scandinavia, Iceland,
northern UK, and some regions of Central and Eastern Europe (Figure 4). The highest values
were predicted in eastern Iberia and many regions of Central and Eastern Europe, although a
big part of the study area was dominated by neutral pH values (6 to 7). The spatial patterns for
GW-Ca (Figure 4) were rather similar to pH. The overall correlation between the two models
was 0.83 (Pearson r, P < 0.001), but they showed differences in some regions. This was
supported by the spatial correlation computed for each cell (Figure 5), reflecting different
magnitudes of correlation across the study area, especially in the eastern Iberian Peninsula and
Southeastern Europe.

**Figure 4**. Model predictions based on Regression Kriging. Note the $Ca^{2+}$ concentration is on ln-
scale.



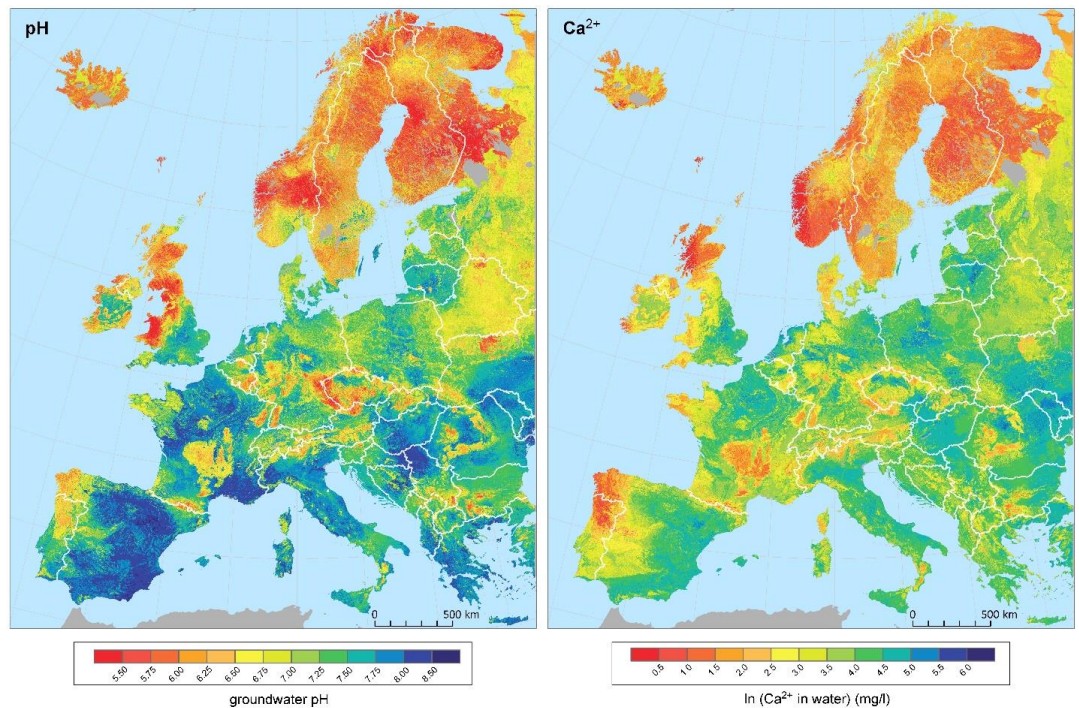





**Figure 5**. Spatial correlation between the models computed for groundwater pH and Ca$^{2+}$.
Values show Pearson correlation coefficient computed over every single cell by using a
sampling of 25 neighboring cells.

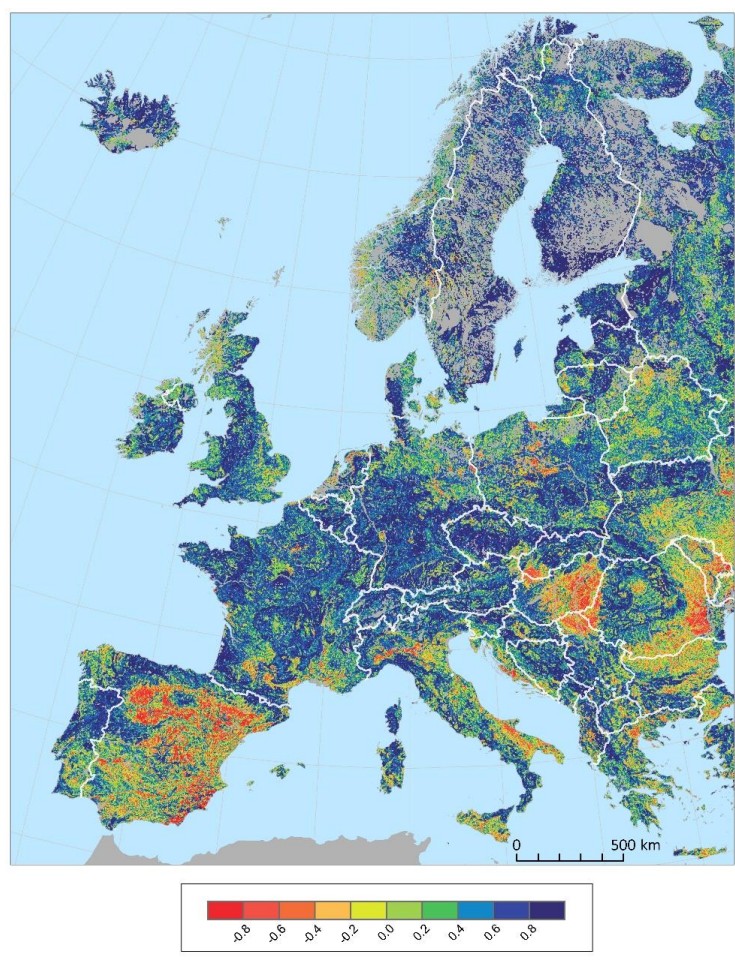


**Discussion**
3.1.  *Spatial patterns in groundwater pH and Ca$^{2+}$ concentration in Europe*
As expected, the recent values of water pH and $Ca^{2+}$ concentration are largely shaped by
lithology in groundwater-dependent habitats across Europe. Indeed, it has been recognised by
regional studies that the distribution of major spring and fen habitats, of which the species
composition largely depends on pH and $Ca^{2+}$, is well determined by bedrock type (Hájek et al.
2002, Tahvanainen 2004, Hinterlang 2017, Peterka et al. 2017, Cantonati et al. 2020c). Since the
European-scale geological map used here is not precise enough to capture differences in
bedrock chemistry within the major geological units that are defined largely by age, the
contribution of soil pH in the model probably also reflected lithological variation, as soil pH
generally correlates with regional bedrock chemistry (Chadwick and Chorover 2001). On the
other hand, soil pH is also affected by climate-dependent pedogenesis, which incorporates a
climate-zonal geographical component in this effect (Duchaufour 2012, Maxbauer et al. 2017).
Precipitation is another important determinant of groundwater chemistry. Precipitation
amount and frequency affect not only flow paths activity and redistribution of groundwater,
but also its residence time in the aquifer, impacting carbonate dissolution and precipitation
rates (Crossman et al. 2011, Lewandowski et al. 2015, Vystavna et al. 2020). Groundwater with
a short transit time (1–3 years) or 'young water' (Soulsby et al. 2015) can be particularly
sensitive to changes in precipitation amount and frequency. Here, intensive precipitation may
reduce an interaction time of groundwater with $Ca^{2+}$ and carbonates deposited in rocks
(Fairchild et al. 1994, Segadelli et al. 2017, Cantonati et al. 2020b), resulting in lower
concentration of elements in groundwater. In snow-influenced ecosystems, seasonal snowmelt
can also modulate the recharge patterns of groundwater. Particularly, the duration of the
snowmelt period can impact the occurrence and dynamic of preferential flow, and prolong or
reduce the interaction of the seepage with soil and bedrock materials (Mohammed et al. 2019).
We therefore suggest that fast hydrological pathways and short transit time driven by
snowmelt and precipitation can explain the lowest $Ca^{2+}$ in hyper-oceanic cold regions of SW
Norway or W Scotland. It may further lower pH and $Ca^{2+}$ values on windward slopes of high
mountains, even if bedrock is moderately calcium-rich.

The resulting pattern at the European scale is governed by the strong latitudinal and

altitudinal gradients, i.e. decreasing pH and $Ca^{2+}$ northwards, and regionally also towards
mountain regions. Although this pattern is well known (Økland et al. 2001, Hájek et al. 2006,
Hinterlang 2017, Peterka et al. 2017) and has been captured also by the FOREGS Geochemical
Atlas of Europe (Salminen et al. 2006), our improved model provides much finer regional
patterns. In Southern Europe, low pH and $Ca^{2+}$ values were modelled in the Pyrenees, the
Balkans, SW Corse, and Calabria, i.e. the regions where boreal or endemic types of fen
communities occur as relicts (Chytrý et al. 2020). The Alps, the Apennines, the Carpathians, and
the Baltic region show a fine-scaled mosaic of alkaline (calcium-rich) and acidic (calcium-poor)
groundwater that contributes to the high diversity and conservation value of groundwater-
dependent ecosystems, such as fens (Cantonati et al. 2009, 2011, Gerdol et al. 2011, Joosten et
al. 2017, Horsáková et al. 2018). The most apparent "acidic island" in Central Europe is located
in the SW part of the Bohemian Massif (Czech Republic, Germany), where acidic types of
springs and fens are quite frequent and some studies further document anthropogenic
acidification on siliceous bedrock in 1970–80s, being re-emerged recently because of extreme
climatic events (Kapfer et al. 2012, Schweiger et al. 2015). It is, however, possible that
particularly this acidic island is picked out mainly because of the high amount of available data.



362 Clearly, most of Fennoscandia is markedly acidic and calcium-poor mainly due to glacial

363 history. Yet, the model identified small alkaline and calcium-enriched islands in NE and Central

364 Sweden, and NW Norway, which are associated with rare types of calcareous fen and spring

365 communities (Dierssen 1982, Vorren et al. 1999, Udd et al. 2015, Miller et al. 2020). More

366 localised pockets of calcareous habitats are however known from most parts of Fennoscandia

367 that are not recognised with the grain of our European-wide analysis. With our results, the

368 future modelling of diversity and distribution of individual habitat types of groundwater-

369 dependent wetlands will be more reliable, regionally rare habitat conditions can be better

370 recognised, and the disentangling of the climate and pH effects will be more easily feasible.

372 3.2. *Persisting data gaps*

373 Although being based on the hitherto most comprehensive field data set, the presented

374 map cannot be considered definitive. Surely there are many pH and EC or $Ca^{2+}$ measurements

375 conducted across Europe that we could not include into the data set because they are hardly

376 accessible. Except for Russian Federation and Moldova, largest gaps still occur in the southern

377 parts of the Pannonian plain (southeastern Hungary, northern Serbia, and western Romania), in

378 SE Belarus, and eastern Ukraine. We dispose of some data from the latter region (Vystavna et

379 al. 2015; Supplementary Table 2), but a large gap in the rest of the data set prevented reliable

380 geospatial modelling. These data might be used in future updates of the map once the gap in

381 Central Ukraine is filled. The lack of data in the Pannonian plain has led to poor correlation

382 between predicted pH and $Ca^{2+}$ values (Fig. 4). Such a poor correlation and sometimes low

383 density of data apply also for some other lowland regions, such as the Danube plain in S

Romania, Po valley in Italy and valleys around the Duero, Ebro, and Tagus rivers in Spain. Apart
from eutrophication, this result may be caused by the imbalanced distribution of groundwater-
dependent habitat types in our data set. Unlike mountain regions, the data for these lowlands
were largely taken from the FOREGS Geochemical Atlas of Europe (Salminen et al. 2006) and
national groundwater databases, i.e. largely from stream water. Considering the major purpose
of our map, the ecological modelling of fens and springs, these regions are generally not as
important because of the low number of existing target habitats, as they have largely been
transformed to arable land or they are too dry. On the other hand, caution is needed when
interpreting the maps in an ecological sense. The extremely high pH (> 8) and $Ca^{2+}$ (ln $[Ca^{2+}]$ > 4;
i.e., $Ca^{2+}$ > 55 mg.l$^{-1}$) values that occur in lowlands visually govern the map, but for ecological
differentiation of groundwater-dependent habitats in Europe the differences within the middle
part of the gradient, i.e. between pH 5.5 and 7.0, are much more important (Malmer 1986,
Wheeler ＆ Proctor 2000, Hájek et al. 2006, Rydin and Jeglum 2013). In any case, our data set is
expected to be amended in the future, as more studies will be published and more data will be
available. New data will help to improve predictions for those regions with relatively lower
sampling effort, and also those with a lithologically heterogeneous landscape. Future updates
of the model may also focus at finer spatial resolution (e.g., 100 to 250 m) but this will require
to increase the spatial accuracy of the calibration data and the predictor variables.

Despite these persisting gaps, our European map of near-surface groundwater pH and

EC provides the best solution for modelling the biodiversity of groundwater-dependent
ecosystems, especially at the continental or supra-regional scale. We even believe that this map





could be more suitable also for ecological modelling of other than groundwater-dependent
habitats. It may mirror the bedrock chemistry better than the map of soil pH, because soil pH is
a resultant of pedogenetic processes, which are tightly associated with the character of the
vegetation cover itself (Miles 1985, Duchaufour 2012).

**Conclusions**
Here, we provide the first European map of groundwater pH and $Ca^{2+}$ content. We collected as
even as possible distributed field measurements of water pH and $Ca^{2+}$ or EC from European
groundwater-dependent habitats, having high data density in regions rich in endangered
groundwater-dependent ecosystems (springs, fens), and used geospatial modelling. The model
considered predominantly lithology and soil pH (i.e., variables surrogating bedrock chemistry)
and precipitation sum (i.e., residence time of groundwater). Our results also provide a freely
accessible map that can be used in any kind of spatial modelling, showing better resolution and
fewer gaps than previously published maps. The character of our input data, which are also
freely accessible, predetermines our map for being used in ecological modelling to address the
distribution and diversity of groundwater-dependent ecosystems and associated species.
Moreover, we assume the map will represent the best choice also for other types of earth
modeling, because unlike previous maps we included Eastern-European and Balkan countries
and considered lithology in geospatial modelling.

**Data availability**




The dataset of georeferenced pH and EC measurements and the resulting maps in GIS-
compatible format (shapefile) are accessible at www.zenodo.org; doi 10.5281/zenodo.4139912
(Hájek et al. 2020).

**Code availability**
No original R code was used; the used codes are cited.

**Sample availability**
No geoscientific samples registered as International Geo Sample Number (IGSN) have been
used for the manuscript.

**Team list**
Michal Hájek (M.Há.), Borja Jiménez-Alfaro (B.J.-A.), Ondřej Hájek, Lisa Brancaleoni (L.B.),
Marco Cantonati (M.Can.), Michele Carbognani (M.Car.), Anita Dedić (A.D.),  Daniel Dítě (D.D.),
Renato Gerdol (R.G.), Petra Hájková (P.H.), Veronika Horsáková (V.H.), Florian Jansen (F.J.),
Jasmina Kamberović (J.Kam.), Jutta Kapfer (J.Kap.), Tiina Kolari (T.K.), Mariusz Lamentowicz
(K.L.), Predrag Lazarević (P.L.), Ermin Mašić (E.M.), Jesper Erenskjold Moeslund (J.E.M.), Aaron
Pérez-Haase (A.P.-H.), Tomáš Peterka (T.P.), Zuzana Plesková (Z.P.), Alessandro Petraglia (A.P.),
Eulàlia Pladevall-Izard (E.P.-I.), Stefano Segadelli (S.S.), Yuliya Semeniuk (Y.S.), Patrícia Singh



(P.S.), Anna Šímová (A.Š.), Eva Šmerdová (E.Š.), Teemu Tahvanainen (T.T.), Marcello Tomaselli
(M.T.), Yuliya Vystavna (Y.V.), Claudia Biţă-Nicolae (C.B.-N.), Michal Horsák (M.Hor.).

**Author contribution**
M.H. and B.J-A. contributed equally to the paper. They conceived the research, collected data
and led writing. B.J-A. designed and performed Random Forest and Regression Kriging models.
O.H. prepared the input data and final map outputs. All authors provided unpublished data and
commented on the manuscript.

**Competing interests**
The authors declare that they have no conflict of interest.

**Acknowledgements**
This work was supported by the Czech Science Foundation [grant numbers 19-01775S (support
for B.J.-A., P.H., V.H., M.Hor.) and GX19-28491X (Centre for European Vegetation Syntheses;
support for M.H., T.P. and O.H.)]. J.Kap. was supported by The Fram Center (grant nr. A36214).
We thank Dr. Sabine Bergmann and Dr. Dirk Hinterlang (State Agency for Nature, Environment
and Consumer Protection North Rhine-Westphalia, Germany), and Mrs Rossitza Gorova
(Ministry of Environment and Water of Bulgaria) for providing us the public water chemistry
data, Dr. Andraž Čarni for alerting us on the open data on Slovenian groundwater and Dr.



Valerijus Rašomavičius for providing us the data from Kadūnas et al. (2017). We thank Tatyana
Ivchenko for providing us the data for Ural Mts which were not finally included into the paper.
Paweł Pawlikowski (P.P.), Lucia Sekulová (L.S.), Jana Navrátilová (J.N.), and Dag-Inge Øien (D.-
I.Ø.) kindly agreed to use their unpublished pH and EC data. We further thank Ilona Knollová,
John Rodwell (J.R.), Kiril Vasilev (K.V.) and Idoia Biurrun (I.B.) for providing pH data from
vegetation databases via European Vegetation Archive.

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
