# Peer review of "A European map of groundwater pH and calcium"

_Earth System Science Data, 2020_

## Referee Comment (RC1) · Anonymous Referee #1 · 8 Jan 2021

General comments

The authors use a suite of statistical model and machine learning approach to develop a European-wide database of groundwater pH and calcium composition from a large-sample of on-site measurements across Europe. These groundwater parameters are major predictors of ecological status and biodiversity of groundwater bodies. This new database provides some improvements over existing similar database and constitutes an important contribution to the scientific community as well as present and future sustenance of groundwater dependent ecosystems.

I believe that the manuscript is suitable for publication. But the authors could provide clarifications on the specific comments raised below:

[Figure]

Specific comments for authors

1. You have performed quality control to filter unfit datasets before the geographical modelling and numerical analysis, yet modeling exercise such as this are associated with prediction uncertainties. Can you provide an estimate of uncertainties associated with the estimated GW-pH and GW-Ca produced by the model?

3. Is it possible to improve the performance of the random forest model? Which additional predicting variable(s) (even if such information is scarce) could be added to improve the results?

3. What are the criteria for selecting the three predictor variables (soil pH, lithology, and precipitation) used as proxies for predicting groundwater pH and Ca concentration?

---

## Referee Comment (RC2) · Anonymous Referee #2 · 12 Jan 2021

General Comments. This paper describes the development of models estimating shallow groundwater Ca concentrations and pH and maps of the predictions made with these models across Europe. Random forest models developed were based on lithology, soil pH, and precipitation. Authors note that improvements can be made, but the data is a significant improvement over current data sources, and is usable for understanding expected water chemistry in shallow groundwater ecosystems. The data clearly meets the requirement of being unique, useful, and complete. There are a few instances where a little more detail or explanation would make the process used and resulting models easier to understand. A bit more discussion of predictors not included in the model would help guide later efforts to build upon this work.

Specific comments Ln 98-100: This may be a good place to also mention the dominant

role lithology plays in determining groundwater chemistry that would also cause soil pH to be a poor predictor of groundwater dependent habitats. Ln 187: May be worth noting that the resulting equation is similar to what was observed elsewhere (e.g., Hem JD. Study and interpretation of the chemical characteristics of natural water. Department of the Interior, US Geological Survey; 1985) Ln 240-241: The decision on which resolution to use could be explained in more detail. Also, how were multiple observations within a single cell handled? Ln 280-285: Partial dependence plots of each variable should be summarized (i.e., state direction of effect) here, and plots themselves included here or in supplemental material. See R package 'pdp'. Ln 329-330: Inferring potential mechanisms could be misleading without knowing direction of effect from a partial dependence plot. Ln 331-335: Although precipitation amount has an effect on residence time, slope also has a large effect and probably should have been considered as a predictor. Dilution of shallow groundwater by increasing precipitation is another more direct potential cause of lower Ca concentrations. Ln 346-347: It seems that this pattern in pH and Ca could be closely related to temperature, with warmer temperatures leading to faster weathering rates (White, A. F., and A. E. Blum (1995), Effects of climate on chemical-weathering in watersheds, Geochim. Cosmochim. Acta, 59, 1729–1747). Ln 400-401: Could call for more work developing high resolution maps of geochemistry like the ones developed for Germany (Le TD, Kattwinkel M, Schützenmeister K, Olson JR, Hawkins CP, Schäfer RB. Predicting current and future background ion concentrations in German surface water under climate change. Philosophical Transactions of the Royal Society B. 2019 Jan 21;374(1764):20180004.)

Technical corrections: Ln 70-73: this sentence is difficult to follow. Ln 378: Not clear what is meant by "dispose".

---

## Author Comment (AC1) · 5 Feb 2021

Reviewer 1

RC: The authors use a suite of statistical model and machine learning approach to develop a European-wide database of groundwater pH and calcium composition from a large sample of on-site measurements across Europe. These groundwater parameters are major predictors of ecological status and biodiversity of groundwater bodies. This new database provides some improvements over existing similar database and constitutes an important contribution to the scientific community as well as present and future sustenance of groundwater dependent ecosystems.

Author comment (AC): We would like to thank Reviewer for the general positive opinion

about our work.

RC: You have performed quality control to filter unfit datasets before the geographical modelling and numerical analysis, yet modeling exercise such as this are associated with prediction uncertainties. Can you provide an estimate of uncertainties associated with the estimated GW-pH and GW-Ca produced by the model?

Author comment (AC): Besides the numerical estimates of model error and mean standard error, it is difficult to address other forms of numerical model uncertainties in our framework, but we have tried to clarify them as much as possible. Firstly, note that MSE is based on 5-fold cross-validation, thus providing a first estimate of model accuracy. In the new version, the Discussion also makes a general assessment of the quality of data input and predictors, as well as their uncertainties (see next points about selection of variables and potential use of others in the future). We also included a concluding remark in the Conclusions to summarize the different sources of uncertainties related to data sources and the modeling framework: "Despite the accuracy of our models, we note that prediction uncertainties may affect the reliability of models computed with both Random Forests and kriging (Hengl et al. 2018; Szatmári & Pásztor 2019). Another source of prediction uncertainty is related to the quality of the original chemical measurements and the georeferentiation of their geographic position. Moreover, the predictor variables rely on spatial models (soil pH, precipitation) or broad geographic maps (lithology) which are based on their own uncertainties and assumptions. Future improvements of GW-pH and GW-Ca estimates should therefore consider a more accurate set of response variables and predictors ,...".

RC: Is it possible to improve the performance of the random forest model? Which additional predicting variable(s) (even if such information is scarce) could be added to improve the results?

Author comment (AC): A significant improvement in the prediction value of the models would need a set of fine-scale lithological (or soil chemistry) maps at the continental

scale, which is not currently available. Although such layers are in the pipeline of geological mapping initiatives, it will take long time before such products will be available, because they also will need the collection of massive field data. Since this is relevant point to understand the current and future versions of our modeling framework, in the version of the Discussion we included a new paragraph to explain predictors that could be included and the lack of suitability of the excluded variables (as suggested by reviewer 2) – please see the last paragraph of the Discussion, before the Conclusions which has been rephrased to address this point.

RC: What are the criteria for selecting the three predictor variables (soil pH, lithology, and precipitation) used as proxies for predicting groundwater pH and Ca concentration?

Author comment (AC): The variables for soil pH and lithology were selected in the exploratory analyses we conducted for every single variable using linear modeling, as explained in the methods (in the section "Numerical analyses"). The inclusion of precipitation was based on the theoretical evidence of relationships between climate and groundwater chemistry. We added this text to the revised manuscript: "We focused on the predictors that may causally affect the groundwater pH and calcium richness. Aquifer chemistry is of prime importance (Hem 1985, Fairchild et al. 1994, Frei et al. 2000, Chapelle et al. 2003, Tahvanainen 2004, Stevens et al. 2020), but no such thematic map exists, at least at the scale needed for computing our spatial predictions. We therefore included the lithological groups from the Hydrogeological Map of Europe (Duscher et al. 2015), together with soil pH maps (see below), for which we anticipated a certain correlation with bedrock chemistry. Apart from aquifer chemistry, residence time may also affect groundwater chemistry by impacting dissolution rates. Precipitation amount and frequency affect not only flow paths activity and redistribution of groundwater, but also its residence time in the aquifer, impacting carbonate dissolution and precipitation rates (Hem 1985, Crossman et al. 2011, Lewandowski et al. 2015, Vystavna et al. 2020). Groundwater with a short transit time (1–3 years) or 'young water' (Soulsby et al. 2015) can be particularly sensitive to changes in precipitation
amount and frequency. We therefore also considered climatic parameters associated with precipitation in the models (see below). Although there are some other potential predictors of minor importance that may affect groundwater chemistry (Hem 1985, Stevens et al. 2020), no corresponding thematic map is available to be included into our models. For some sites, the selected predictors were missing in the maps (e.g., sites at far north in the arctic zone, or close to sea or water bodies) and these sites were therefore not included in the final models. We finally collected topographic data to test the potential effect of elevation and slope as indirect factors potentially influencing groundwater chemistry.".

Reviewer 2

RC: This paper describes the development of models estimating shallow groundwater Ca concentrations and pH and maps of the predictions made with these models across Europe. Random forest models developed were based on lithology, soil pH, and precipitation. Authors note that improvements can be made, but the data is a significant improvement over current data sources, and is usable for understanding expected water chemistry in shallow groundwater ecosystems. The data clearly meets the requirement of being unique, useful, and complete. There are a few instances where a little more detail or explanation would make the process used and resulting models easier to understand.

Author comment (AC): We would like to thank Reviewer for the general positive comment on our work.

RC: A bit more discussion of predictors not included in the model would help guide later efforts to build upon this work.

Author comment (AC): We also considered this is an important issue. In the revised version we included more details about the selection of variables and also about not-included variables and the potential use of new ones in the future – please see the last paragraph of the Discussion, before the Discussion, which has been rephrased to
address this point; some additional sentences are now also in Methods (see previous point for Rev-1) .

RC: Ln 98-100: This may be a good place to also mention the dominant role lithology plays in determining groundwater chemistry that would also cause soil pH to be a poor predictor of groundwater dependent habitats.

Author comment (AC): We added: "Ideally, such models should include lithology as a dominant factor determining groundwater pH and Ca2+ concentration (Hem 1985, Chapelle 2003, Tahvanainen 2004, Stevens et al. 2020)".

RC: Ln 187: May be worth noting that the resulting equation is similar to what was observed elsewhere (e.g., Hem JD. Study and interpretation of the chemical characteristics of natural water. Department of the Interior, US Geological Survey; 1985)

Author comment (AC): We added this reference to the "well-known correlation between EC and Ca2+ concentration" and to other places. Then we explicitly state that: ". . . the resulting equation y = 0.153x was obtained (R2 = 0.84). Such relationship between Ca2+ and EC is similar as that found in the abovementioned studies (Hem 1985, Sjörs ïijĘ Gunnarsson 2002, Plesková et al. 2016)".

RC: Ln 240-241: The decision on which resolution to use could be explained in more detail. Also, how were multiple observations within a single cell handled?

Author comment (AC): This is indeed an important point to remark. The revised version says: "The variables of lithology and soil pH were aggregated to the same grid extent of CHELSA at 1 km resolution, as the most appropriate scale to balance the original scales of both layers. This grid extent is also the most suitable spatial scale to be used in the context of further ecological modeling, which is in many cases combined with climatic data from e.g. CHELSA or WorldClim (www.worldclim.org) for making temporal climatic projections. The lithological map (originally at 1:1,500,000 scale, which corresponds to a raster resolution of c. 1 km) was converted to a grid resolution using

the dominant unit. Soil pH was converted from the original 250 m to 1 km grid resolution using a bilinear interpolation to create a smooth surface based on the weighted average of the four nearest cells".

RC: Ln 280-285: Partial dependence plots of each variable should be summarized (i.e., state direction of effect) here, and plots themselves included here or in supplemental material. See R package 'pdp'.

Author comment (AC): Thanks for this remark. We have produced the partial effect plots and they are now included in the main text as Figure 4. The revised version of Results includes details about these effects, and they are also used for the Discussion.

RC: Ln 329-330: Inferring potential mechanisms could be misleading without knowing direction of effect from a partial dependence plot.

Author comment (AC): We rephrased the text according to the dependence plots.

RC: Ln 331-335: Although precipitation amount has an effect on residence time, slope also has a large effect and probably should have been considered as a predictor. Dilution of shallow groundwater by increasing precipitation is another more direct potential cause of lower Ca concentrations.

Author comment (AC): Yes, we already mentioned the effect of precipitation in the first version of manuscript, but in the revised manuscript we stress this effect even more. Concerning the effect of slope on residence time, it is the only comment we did not addressed in the text. In our opinion, topography has a stronger effect on the surface water residence time. In the groundwater system, the flow path length is a more important control on residence times than the hillslope gradient. Therefore, we have a doubt about the slope as a predictor of the groundwater residence time at the larger scale; see also: doi:10.1029/2006WR005393. In any case, our exploratory analyses reflected a negligible effect of slope, we added: "Slope and elevation showed negligible effects on both GW-pH and GW-Ca ((linear regression, adjR2 < 0.05, P < 0.001". In the

[Figure]

Discussion, however, we recognize that further models at better resolution may provide stronger effects, thus we added: "Nevertheless, understanding the complementary effects of precipitation and slope will need to account for more accurate models based on GPS data, better precipitation data, and high-resolution (<250 m) topographic predictors"

RC: Ln 346-347: It seems that this pattern in pH and Ca could be closely related to temperature, with warmer temperatures leading to faster weathering rates (White, A. F., and A. E. Blum (1995), Effects of climate on chemical-weathering in watersheds, Geochim. Cosmochim. Acta, 59, 1729–1747).

Author comment (AC): We changed this part as follows: "The resulting pattern at the European scale is governed by the strong latitudinal and altitudinal gradients, i.e. decreasing pH and Ca2+ northwards, and regionally also towards mountain regions. This pattern largely follows bedrock chemistry, with crystalline rocks prevailing, and most carbonate rocks being eroded by glaciers, in high latitudes and altitudes. The excess of precipitation over evaporation, and theoretically also slower weathering rates in colder regions (White ïijĘ Blum 1995), contribute as well".

RC: Ln 400-401: Could call for more work developing high resolution maps of geochemistry like the ones developed for Germany (Le TD, Kattwinkel M, Schützenmeister K, Olson JR, Hawkins CP, Schäfer RB. Predicting current and future background ion concentrations in German surface water under climate change. Philosophical Transactions of the Royal Society B. 2019 Jan 21;374(1764):20180004.)

Author comment (AC): We added: "...as in some Central-European areas (Le et al. 2019, Chuman et al. 2019)".

RC: Ln 70-73: this sentence is difficult to follow.

Author comment (AC): We simplified this sentence: "For ecological models of distribution and diversity of many terrestrial ecosystems, the new map is more suitable

than maps of soil pH, which mirror not only bedrock chemistry, but also vegetation-dependent soil processes".

RC: Ln 378: Not clear what is meant by "dispose". Author comment (AC): corrected